**Data Availability Statement:** The data underlying the results presented in the study are available from the Ministry of Health, Community Development, Gender, Elderly and Children through

# Social accountability in primary health care facilities in Tanzania: Results from Star Rating Assessment

Erick S. Kinyenje[1]*, Talhiya A. Yahya[1], Joseph C. Hokororo[1], Eliudi S. Eliakimu[1], Mohamed A. Mohamed[2,3], Mbwana M. Degeh[1], Omary A. Nassoro[1], Chrisogone C. German[1], Radenta P. Bahegwa[1], Yohanes S. Msigwa[1], Ruth R. Ngowi[1], Laura E. Marandu[1], Syabo M. Mwaisengela[4]

1 Health Quality Assurance Unit, Ministry of Health, Community Development, Gender, Elderly, and Children, Dodoma, Tanzania, 2 Tanzania Field Epidemiology and Laboratory Training Programme (TFELTP), Dar es Salaam, Tanzania, 3 East Central and Southern Africa Health Community, Arusha, United Republic of Tanzania, 4 Regional Administrative Secretary's Office—Regional Health Management Team, Mtwara, Tanzania

* kinyenje2003@yahoo.com

## Abstract

### Background

Star Rating Assessment (SRA) was initiated in 2015 in Tanzania aiming at improving the quality of services provided in Primary Healthcare (PHC) facilities. Social accountability (SA) is among the 12 assessment areas of SRA tools. We aimed to assess the SA performance and its predictors among PHC facilities in Tanzania based on findings of a nationwide reassessment conducted in 2017/18.

### Methods

We used the SRA database with results of 2017/2018 to perform a cross-sectional secondary data analysis on SA dataset. We used proportions to determine the performance of the following five SA indicators: functional committees/boards, display of information on available resources, addressing local concerns, health workers' engagement with local community, and involvement of community in facility planning process. A facility needed four indicators to be qualified as socially accountable. Adjusted odds ratios (AOR) with 95% confidence intervals (CI) were used to determine facilities characteristics associated with SA, namely location (urban or rural), ownership (private or public) and level of service (hospital, health centre or dispensary).

### Results

We included a total of 3,032 PHC facilities of which majority were dispensaries (86.4%), public-owned (76.3%), and located in rural areas (76.0%). On average, 30.4% of the facilities were socially accountable; 72.0% engaged with local communities; and 65.5% involved communities in facility planning process. Nevertheless, as few as 22.5% had functional Health Committees/Boards. A facility was likely to be socially-accountable if public-owned

Tel: +255-22-2342000/5 Email: ps@afya.go.tz.
Data are third party. Most of the authors belong to
the "user department" i.e. the department of Health
Quality Assurance that uses the said data for day-
to-day quality improvement. Therefore, we are
granted special access to data by the Ministry of
Health.

**Funding:** The author(s) received no specific
funding for this work.

**Competing interests:** The authors have declared
that no competing interests exist.

[AOR 5.92; CI: 4.48–7.82, p = 0.001], based in urban areas [AOR 1.25; 95% CI: 1.01–1.53, p = 0.038] or operates at a level higher than Dispensaries (Health centre or Hospital levels)

## Conclusion

Most of the Tanzanian PHC facilities are not socially accountable and therefore much effort in improving the situation should be done. The efforts should target the lower-level facilities, private-owned and rural-based PHC facilities. Regional authorities must capacitate facility committees/boards and ensure guidelines on SA are followed.

## Introduction

Quality of health care can be defined as the delivery of health services that are effective, safe and patient-centred, delivered in a way that is timely, equitable, integrated and efficient [1]. In the global efforts to ensure attainment of the sustainable development goal 3 (good health and wellbeing)–especially target 3.8 (achieve universal health coverage—including financial risk protection, access to quality essential healthcare services and access to essential medicines and vaccines for all), countries need to ensure that their health systems are providing high-quality health care services. High-quality health care services refer to: "*the right care, at the right time, in a coordinated way, responding to the service users' needs and preferences, while minimizing harm and resource waste*" [2] Quality of health care in Tanzania faces several problems including inadequate supportive supervision in health facilities by management teams in Local Government Authorities; lack of ownership of quality improvement at the facility level; inadequate implementation of infection prevention and control measures including health care waste management [3]. Other problems are inadequate implementation of water, sanitation and hygiene standards; breach of ethics and professional conduct by health workers; low motivation of health workers; and inadequate compliance to guidelines and standards by health care workers [3]. Achieving sustained QI requires commitment from the entire organization, particularly from top-level management. In view of that the Ministry of Health in Tanzania in collaboration with the President's Office—Regional Administration and Local Government and other stakeholders during the design of big-results now initiative in the health sector, four interventions were identified in which one of them was the performance of primary health care (PHC) facilities; in which one of the activities was the quality assessment of all PHC facilities [4]. To achieve this, the stakeholders looked at a variety of existing approaches or QI models, such as improvement collaborative [5]; step-wise certification towards accreditation using safe care standards [6] and electronic supportive supervision tool for primary health facilities [7]. They also looked at supervision and mentorship tool for HIV and AIDS services [8]; and continuous quality improvement using 5S-(Sort, Set, Shine, Standardize, Sustain) approach [9]; in order to choose the best approach that will help to collect and analyse data and test change in the quality of services provided in primary health care (PHC) facilities. In the end, the ministry had used a model of stepwise improvement process towards a pre-accreditation status (star level 5) known as Star Rating for health facilities with the vision to increase the effectiveness of QI in healthcare which was conducted in 2015/2016 (as baseline) and reassessment done in 2017/2018 [4]

The SRA initiative aimed at assessing all the PHC facilities across the country and assigning a star level according to the standard of services provided based on a set of tools for dispensaries, health centres, and level 1 hospitals [4]. According to the health system of Tanzania, the PHC facilities are those providing services at lower levels with no speciality expertise level.

They include dispensaries at the village level, health centres at ward level and hospital level 1 at council/district level. At speciality expertise level (referral level services) include hospitals level 2 at regional, level 3 at zonal and level 4 at national level [10, 11]. The SRA tools are arranged into 12 service areas, which are: Legality (Licensing and Certification), Health Facility Management, Use of Facility Data for Planning and Service Improvement, Staff Performance Assessment, Organization of Services, Handling Emergencies and Referral, Client Focus, Social Accountability, Facility Infrastructure, Infection Prevention and Control (IPC), Clinical Services, and Clinical Support Services [4, 12].

In this paper, we describe the status of implementation of "social accountability" which is service area eight in the SRA tools. In the context of PHC, social accountability is a measure of whether a country and especially the health facility, are held accountable to existing and emerging social concerns and priorities based on need [13]. Social accountability strategies "*try to improve institutional performance by bolstering both citizen engagement and the public responsiveness of states and corporations*" [13]. Social accountability offers a set of approaches and tools to promote citizen engagement and monitoring to improve system performance, effectiveness, and responsiveness to public needs. Because different countries, regions, or even communities face different breakdowns in PHC, this set of approaches provides a mechanism for citizens and civil society, together with service providers and government, to identify and seek solutions to specific problems they observe with their local health system. Effective social accountability is enabled through regular feedback loops between health system users and administrators [13]. During the SRA, the following five indicators were being assessed: Healthcare workers engagement with the local community; facility addressing local concerns; community participation in the facility planning process; displaying key information on available resources; and Health Facility Governing Committee (HFGC) or Health Facility Board (HFB) activeness and well oriented to provide feedback to the broader community.

The issue of accountability in health systems has been part and parcel of the health sector reforms globally [14]. In sub-Saharan African countries, emphasis on accountability in terms of citizen participation in decision making in the health sector was cemented by Health Ministers in 1987 in Bamako, Mali in a conference that came with what is known as the "Bamako Initiative", which had several principles to adhere to including "*public participation in decision-making, and decentralized implementation of programmes at the level of the district health system*" [15]. The initiative aimed to help sub-Saharan African Countries to strengthen PHC services amid the economic crises that affected social services. The Bamako Initiative came 3 years after the Local Government Authorities in Tanzania were re-established in 1984 following the passage of legislation in 1982.And as part of strengthening the local governments, from the mid-1990s, the government started to implement the "Local Government Reform Programme" that had six components, one of them being "governance" which aimed at "*establishing a broad-based community awareness of participation in the reform process and promote principles of democracy, transparency and accountability*" [16, 17]. Also, the first National Health Policy of 1990, emphasized community participation and having full say about their health as a pre-requisite for implementation of PHC [18].

Also, as part of the wider civil service/public sector reforms in Tanzania [19]; the health sector also underwent reforms, which included the development of a proposal for health sector reforms in 1994 [20]. Implementation of decentralization of health sector to local governments authorities (also known as councils) has had several benefits including strengthening of health workers' accountability, but with some challenges in some areas where "*lack of community participation in planning*" has been reported [21]. At the council level, the avenue for citizens to voice their needs and expectations as well as participate in planning is through the HFGC for dispensary and health centre and HFB for level 1 hospital. Composition of HFGC/HFB

includes: three (3) members from the population served by the respective facility, one member from faith-based organizations, and one member from private for-profit institutions; and their selection is done transparently at the level of their authority. To ensure gender representation, at least one-third of members must be women. Also, the HFGC/HFB are accountable to the Council Health Services Board [22]. The guidance on the selection of members of the HFGC/HFB provides clear linkages with other structures in the council, reporting channels, the knowledge required, and involvement of stakeholders, hence supporting the conceptual framework by Molyneux, *et al.* 2012 [23]. The composition of HFGC/HFB is an important element in ensuring social accountability in PHC facilities as noted by Lodenstein, *et al.* 2017 [24].

Health facility committees in other countries have also been shown to play a significant role in improving social accountability in PHC facilities. For example in Malawi, the committees work together with health facility staff by managing social relations around the facility, promoting minimum level of access and quality of services, as well as reporting serious misconducts to health authorities [25]. In West Africa (Benin and Guinea) and Central Africa (Democratic Republic of Congo) the committees ensure social accountability through engagement with health providers in person or through meetings to service failures [24]. A systematic review of the social accountability process in the health sector in sub-Saharan Africa by Danhoundo, *et al.* 2018, has identified several barriers to effective implementation including "*health system barriers, corruption, fear of reprisal, and limited funding*" [26].

This paper aims at assessing the social accountability of public and private PHC facilities in Tanzania as part of the SRA re-assessment that was conducted in 2017/2018. This was part of the then broad government initiative termed "Big Results Now" [4]. The analysis also aims at showing the potential of the SRA Tools to assist as a mechanism for making facility in-charges and other staff accountable for providing quality services. The specific objectives of the study were as follows;

1. To determine the proportion of PHC facilities with functional social accountability mechanisms based on the SRA results.

2. To determine PHC facility characteristics associated with functional social accountability mechanisms based on the SRA results.

## Methods

### Conceptual framework

The assessment components for social accountability including indicators and verification criteria are shown in Table 1. Table 1 was derived from the SRA Tool and modified to a language of publication however, none of the indicators were changed. Several conceptual frameworks looking at various aspects of social accountability have been developed by McCoy, *et al.* 2012 [27]; Molyneux, *et al.* 2012 [23]; Lodenstein, *et al.* 2013 [28]; Lodenstein, *et al.* 2017 [29]; Lodenstein, *et al.* 2017 [24]; Paschke, et *al.* 2018 and Vian, T., 2020 [30] We adapted the frameworks by McCoy, *et al.* 2012 [27]; Lodenstein, *et al.* 2017 [29]; Paschke, et *al.* 2018 [31]; and Vian, T., 2020 [30], and conceptualized that functionality of social accountability in PHC facilities is a combination of the following mechanisms: health workers engagement with the local community, facility addresses local concerns, transparency, the functionality of health facility governing committee/board, and participation as shown in Fig 1.

### Study design

We performed cross-sectional secondary data analysis of the social accountability dataset found in the National SRA database of 2017/2018.

**Table 1. A section on SRA tool assessing social accountability performance at healthcare facilities in Tanzania.**

| Indicator | Definition and verification criteria | Allocated score |
|---|---|---|
| **Functional facility governance committees or boards** | The functional facility governance boards/committees were expected to have the following six characteristics:<br>1. There is up to date list of board members including their contact information.<br>*Verification*: A list was verified from the health facility records<br>2. If board/committee members attend meetings<br>*Verification*: minutes over the past 6 months were checked to see whether the meetings were held with 6 or more members attending (quorum)<br>3. If members had adequately trained and oriented on their roles and responsibilities<br>*Verification*: Reports on training or orientation were checked to confirm that roles and responsibilities of HFGC /HFB were adequately covered. Member of the board were interviewed whenever possible.<br>4. Local concerns, issues, or complaints conveyed through the board.<br>*Verification criteria*: minutes of the board were checked to check whether issues from community were discussed<br>5. If the board held responsible parties accountable in following up the community concerns.<br>*Verification*: minutes of the board were checked to see whether actions were taken to address community complaints raised previously, through matters arising and monitoring of implementation (Any from last 12 months)<br>6. If the board gave feedback to the village/ward social service committee or village/ward assembly.<br>*Verification criteria*: Minutes of village/ward/ social service committee or assembly (any from last 6 months) were checked. | Yes = 1 was awarded to a health facility that scored yes to all 6 questions; No = 0 was awarded if the facility scored less than 6 questions |
| **Key information on available resources is displayed** | If the following information were displayed at facility:-<br>a) Plans and budget<br>b) Allocation of medicines & Supplies<br>c) Revenue collection, received funds and expenditure<br>*Verification criteria*: the above information were checked if could be viewed by the public. | Yes = 1 was given if all 3 items displayed.<br>Partial = 0.5 was given if 2 items displayed.<br>No = 0 was given if less than 2 items displayed |
| **The facility addressed local concerns** | Did the facility management team plan specific interventions to address local health concerns and improve services?<br>*Verification criteria*: specific health facility plans were checked to verify interventions which addressed local community concerns related to health care delivery. | Yes = 1 was given if Facility plan showed interventions/steps to address local health problems identified from the local community; otherwise, No = 0 was awarded |
| **Healthcare workers engage with local community** | Are healthcare workers seen to be engaged with local community concerns related to health care delivery?<br>*Verification criteria*: Check attendance of local/village meetings, (including social service committee meeting). Either Village Executive Officers were interviewed or minutes of village meeting or community meetings were checked to verify attendance of health worker in the past 6 months. | Yes = 1 was given if Local community acknowledged health care workers' engagement, and meeting attendance held in the past 6 months; otherwise No = 0 score awarded. |
| **Community participates in facility planning process** | Is the community engaged during the process of annual planning by the facility?<br>*Verification criteria*: Minutes from facility meetings for preparation of Health facilities' annual plans were checked to verify attendance of community member (s) e.g. member from HFGC/VEO, Village chairperson | Yes = 1 was given if Minutes of meetings showed participation of community member; otherwise No = 0 was given. |

## Study population

SRA data were collected from PHC facilities; the facilities which are responsible for the provision of PHC services in Tanzania. Dispensaries are the lowest level in PHC facilities that provide exclusively outpatients' services to approximately 10,000 population while health centres are designated referral points for dispensaries. Health centres provide a broader range of services including inpatient services and Comprehensive Emergency Obstetric and Newborn

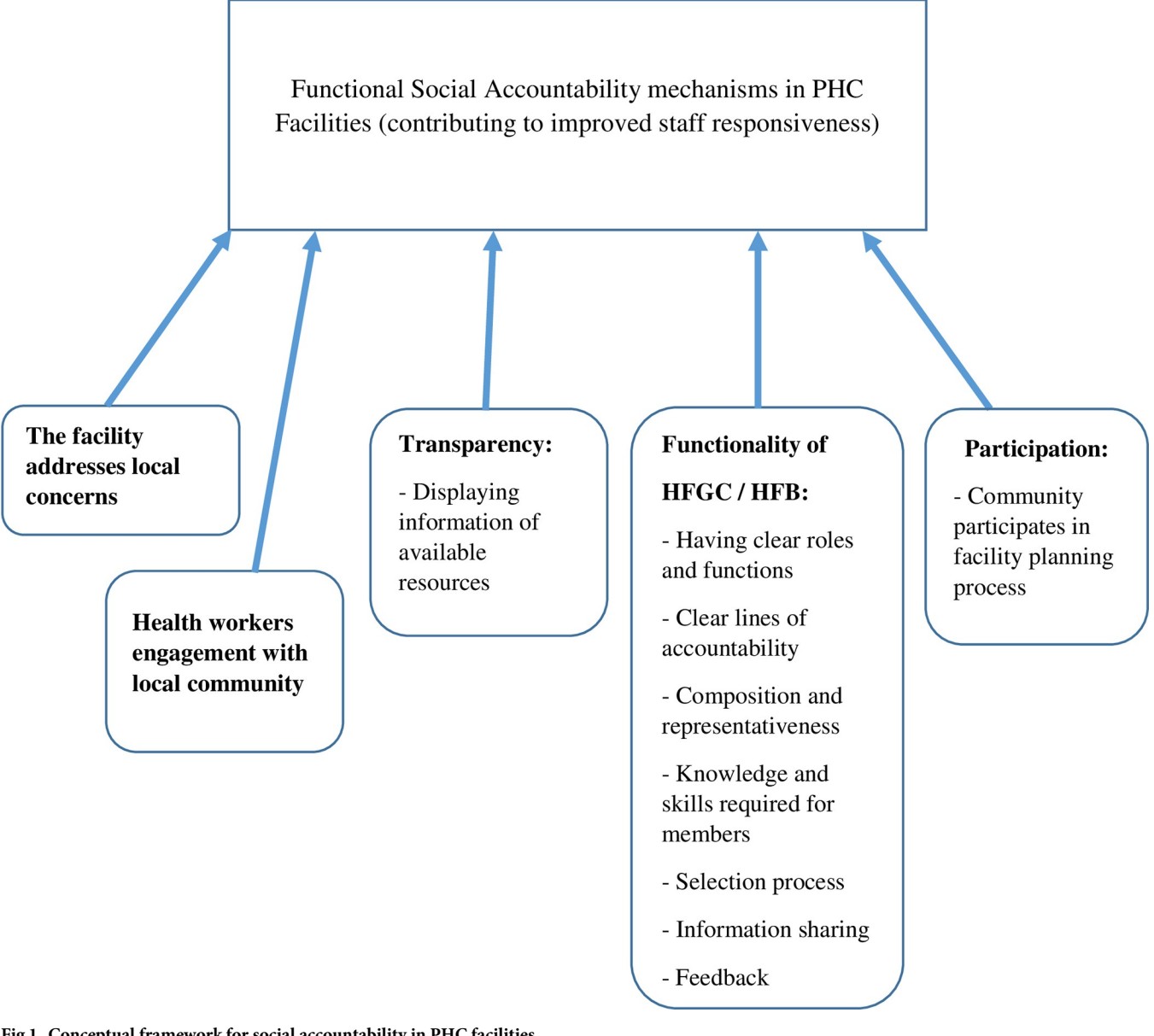

**Fig 1. Conceptual framework for social accountability in PHC facilities.**

Care (CEmONC) to about 50,000 population. A hospital at the council level (i.e., level 1 hospital) serves about 250,000 population and receives referrals from the low levels [32].

There are 184 local government authorities (councils) in Tanzania and each has several public and private-owned health centres and dispensaries and one publicly owned council hospital (or designated private hospital whenever there is no public one). The councils are either located in rural or urban areas with relatively different cultures and socio-economic activities and status.

**Sampling.** All facilities that participated in the 2017/2018 performance assessment.

**Inclusion criteria.** All health care facilities that participated during the 2017/18 assessment were included in this study.

**Exclusion criteria.** The facilities whose performance and characteristics were not identified from the SRA database.

**Star Rating Assessment database.** The Health Quality Assurance Unit (HQAU) of the Ministry of Health manages the data that were collected at two-point national-wide assessments. The database is made of 12 service areas whose performance results are kept in; including the social accountability results. Since the dataset for 2015/16 were mostly incomplete, we used 2017/18 dataset results for 2017/18 for this study.

The section on social accountability is grouped into five indicators namely; Functional facility governance committees or boards, Facility addressed local concerns, Facility addressed local concerns, Healthcare workers engaging with the local community, and Community Participation in the facility planning process. Table 1 presents questions and assessment criteria that were used to score the above five indicators during SRA. For each indicator, there were two available scoring options; Yes' (score = 1) or 'No' (score = 0) except for indicator number 2 "Key information on available resources is displayed" which had the addition of 'Partial' (score = 0.5).

**Data extraction and management.** All social accountability data for the year 2017/18 were extracted and checked for quality and the missing data were excluded in analysis (S1 Appendix). We determined scores for individual indicators and total scores for the area. First, the individual scores were calculated (the score for the indicator divided by the maximum possible score x 100 to give a percentage score). Secondly, the total score across the 5 indicators was determined by calculating the average of the percentage scores for the 5 indicators [33]

## Study variables

The main dependent variable of interest for this study was social accountability. A facility could gain 5 points maximum and they needed 4 to be qualified as socially accountable. This cut-off point (which is equivalent to 80% score) is provided in the National Guidelines for Recognition of Implementation Status of Quality Improvement Initiatives in Health Facilities [34]. The indicator variables outlined in the previous sections were presented as proportions.

Facility's characteristics such as location (rural or urban), health facility level (dispensary, health centre or hospital level 1) and health facility ownership (public or private) were the additional variables that were used to determine association between them and social accountability.

## Data analysis

All analyses were performed using Stata 15. We did categorization and recoding of different variables, and then frequencies and proportions for categorical variables were reported using cross-tabulation tables.

Furthermore, we created a binary variable based on the scores in social accountability which were used to determine an association between the facilities' social accountability and independent variables. The association was measured by calculating the odds ratio with a 95% confidence interval and a $P$-value of $< 0.05$ was considered as statistically significant.

## Results

### Description of participating health facilities

Among 7,289 PHC facilities that were involved in SRA assessment in 2017/2018, 3,032 (41.6%) met inclusion criteria and were eligible for analysis. Table 2 shows that most facilities (86.4%) were dispensaries, public health facilities (76.3%) based in rural areas (76.0%).

**Table 2. Characteristics of health facilities involved in the study (N = 3,032).**

| Variable | Number of HFs (n) | Percent (%) |
|---|---|---|
| **Health Facility level:** | | |
| Dispensaries | 2,615 | 86.42 |
| Health Centres | 311 | 10.28 |
| Hospitals | 100 | 3.30 |
| **Health Facility ownership:** | | |
| Public | 2,306 | 76.31 |
| Private | 716 | 23.69 |
| **Health Facility location:** | | |
| Urban | 727 | 23.98 |
| Rural | 2,305 | 76.02 |

## The proportion of health facilities with functional social accountability mechanisms

Overall, 30.4% (922) of the PHC facilities were found socially accountable (i.e. had at least four out of five functional SA indicators). The average score in percentages of the five indicators was 50.5%; this means that facilities' overall score for performance of social accountability across the five indicators was 50.5%. As it is shown in Fig 2; "facility engagement with the local community" was the most adhered indicator by 72% of the facilities; while only 22.5% of the facilities had functional facility governing committees or boards.

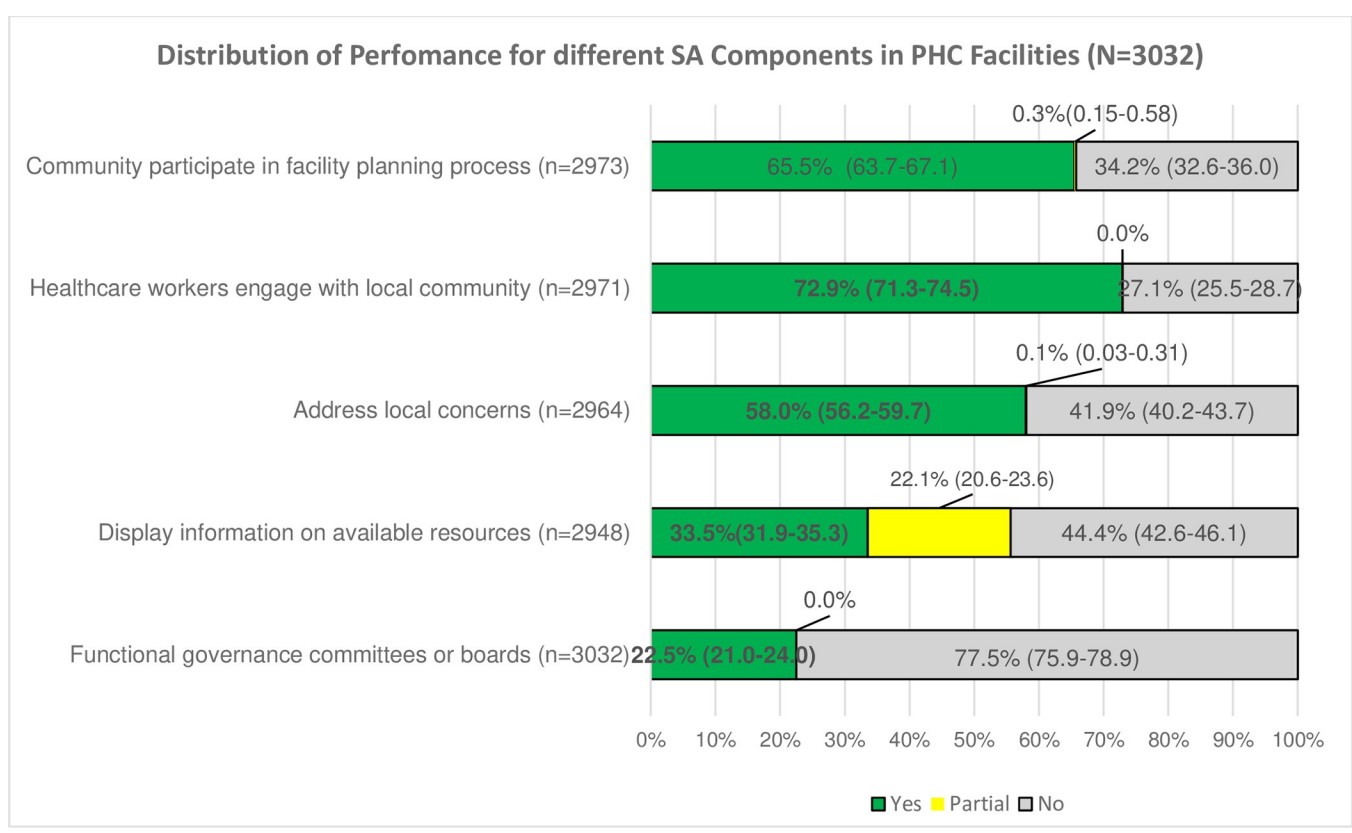

**Fig 2. The distribution of performance for different social accountability components (N = 3032).**

**Table 3. Health facility characteristics associated with social accountability status during Star Rating Assessment of 2017/18.**

| Variable | Socially accountable? | | | | Bivariate | | | Multivariable | | |
|---|---|---|---|---|---|---|---|---|---|---|
| | No | % | Yes | % | AOR | 95% CI | p-value | AOR | 95% CI | p-value |
| **Facility type** | | | | | | | | | | |
| Dispensary | 1,824 | 69.8 | 791 | 30.2 | Ref | | | Ref | | |
| Health Centre | 208 | 66.9 | 103 | 33.1 | 1.14 | 0.89–1.47 | 0.299 | 1.33 | 1.02–1.73 | 0.036* |
| Hospital | 72 | 72.0 | 28 | 28.0 | 0.90 | 0.58–1.40 | 0.631 | 1.94 | 1.18–3.18 | 0.009* |
| **Ownership** | | | | | | | | | | |
| Private | 650 | 90.8 | 66 | 9.2 | Ref | | | Ref | | |
| Public | 1,454 | 63.0 | 852 | 37.0 | 5.77 | 4.42–7.54 | 0.001 | 5.92 | 4.48–7.82 | 0.001* |
| **Location** | | | | | | | | | | |
| Rural | 1,548 | 67.2 | 757 | 32.8 | Ref | | | Ref | | |
| Urban | 562 | 77.3 | 165 | 22.7 | 1.67 | 1.37–2.02 | 0.001 | 1.25 | 1.01–1.53 | 0.038* |

p- Values are calculated using chi-square test.

*Factors whose association were found significant in the final logistic regression model.

Facility type, ownership and location were the variables used to adjust for the association.

### Facility related characteristics associated with social accountability

As shown in Table 3, the odds of being socially accountable were six times among public-owned facilities compared to facilities that are privately owned [AOR 5.92; CI: 4.48–7.82 p = 0.001].

Compared to dispensaries, health centres and hospitals had an increased likelihood of performing well on social accountability by 33% and 94% respectively. PHC facilities that are based in urban areas were likely to be socially accountable compared to rural-based facilities [AOR 1.25; CI: 1.01–1.53, p = 0.038].

## Discussion

This study had focused on the level of social accountability among health facilities in Tanzania and determinants that affect it. Various scholars in sub-Saharan Africa have assessed performance in social accountability among health facilities using different approaches. Mostly they used the performance of health facility governing committees as an indicator of the facility's accountability to society [24, 35–41] while others have used health facility charter [42], citizen report cards [43, 44], and scorecards [45–47]. Our study findings are congruent to a study by Damian has shown existence of poor social accountability among health care facilities [48]. A detailed discussion on the performance of individual indicators plus predictors of SA is following in the next paragraphs.

### Health facility governing committees

The indicator on the functionality of HFGC scored the lowest among the five social accountability indicators. These committees are the instruments to facilitate community participation in the management of human, financial, and material resources needed to provide quality of care in low- and middle-income countries like Tanzania [24, 27]. So far, the evidence from Nigeria, Bolivia and Pakistan suggests that; if HFGCs are able to hold healthcare staff accountable and hence improve quality of care provided that the committees are oriented on their tasks and provided with power [49]. The findings from a neighboring country, Uganda, emphasize that HFGCs' participation alone cannot be productive if members are not well informed [35].

Recent studies from Tanzania show that limited training or orientation among members on their roles is the reason for the poor functionality of HFGCs [50, 51]. In recent years, Tanzania has been emphasizing on Decentralization-by-Devolution (D-by-D) approach whereby HFGCs are provided with more autonomy to govern the PHC facilities towards improved healthcare delivery [21]. However, the D-by-D that is currently implemented has not enabled many committees to have full autonomy and therefore matters pertaining to facilities are still being decided at the highest levels of the country. For example; it is still difficult for the committee to hold the healthcare provider responsible for the misconduct. The professional bodies at national level are the one responsible to investigate such incidences. This is an example of the many potential causes of HFGCs not performing well in Tanzania [48, 52].The voices from Scholars argue the country to implement the true D-by-D to improve the efficiency of HFGCs [48, 53].

Maluka and his colleagues [50] conducted a study in three regions of Tanzania and observed the community was mostly unaware of issues related to operations conducted in the facilities falling in their territories. Lack of community awareness may be a sign that HFGCs were not providing feedback to the village/ward social service committee or village/ward assemblies as required by both D-by-D guidelines and the SRA tool that was used to collect data for this study.

## The displayed information on available resources

Only one-third of facilities displayed information on available resources and this was the second worst-performing indicator of social accountability in our study findings. Resources included in the SRA tool were plans and budget, allocation to medicines and supplies, revenue collection, received funds, and expenditure. It is the requirement that facilities display information relating to facility management on public viewing platforms such as notice boards [54]. Our study did not explore why most facilities performed poorly; nevertheless, the study by Anasel *et al.* (2019) that found similar findings in three regions of Tanzania; documented insufficient skills in data analysis, and the feeling that data are collected for submitting to higher authorities as to the major barriers [54]. The display of information for community consumption is a key to effective social accountability [26], and therefore facilities would improve accountability to society through the provision of a forum for discussing the collected data, making follow-up of complaints, and then provision of feedback to the community [24].

## Health workers' engagement with the local community

Our findings show that health workers engaged with the local community in about three-quarters of the facilities assessed. The engagement was assessed by cross-checking of villages' meetings minutes and then affirmed by villages' leaders. Tanzania has been very successful in the provision of outreach healthcare services at the community level [55, 56], the services whereby health workers are given opportunity to convene community-based meetings and inform the public about the services they will provide for the specific locality and time. From these meetings, minutes are prepared and kept in village administrative offices. Country's high achievement in community healthcare outreach services could have resulted in good performance in community engagement.

## Engaging community during the process of annual planning

Two-thirds of health facilities had functional mechanisms that engage the community during the process of annual planning. This high score could be attributed to the implementation of the decentralisation policy which started about two years before the collection of data that were used for this study. Decentralization requires the involvement of HFGCs during

planning. Our findings reveal an improved situation in the country when compared to the period in which the implementation of this policy had not begun, a time in which community representatives were hardly involved in health facility financial planning [48].

Recent studies suggest that community participation towards improving social accountability at health facilities is hampered by manpower, finance, and infrastructural deficits [57, 58]. In the Tanzanian context, the above challenges may lead to difficulties in achieving these meetings on time because of staff shortage to administer the meetings, inadequate funding needed to cover costs incurred by participants to attend meetings plus conference packages that include stationeries, refreshments and venue. Apart from the above managerial challenges, we believe that community participation during the process of annual planning in Tanzania was mainly challenged by inadequate awareness of rights and responsibilities among communities an explanation which is supported by findings from other previous Tanzanian studies [59, 60].

### Facility related characteristics associated with social accountability

Wangui Machira [61] argues that social accountability is influenced by location and as result, the performance of health facilities is attributed to a range of economic, social, and physical diversity. We suggest urban-based facilities in Tanzania are relatively more equipped with resources such as human capacity and financial resources that are needed for the implementation of components of SA.

Research findings suggest that citizens from rural areas are relatively less educated [49], lacking interest and are having limited to access information [37] and therefore are less likely to participate in social accountability activities compared to urban-based citizens. In South Africa, HFGCs in rural areas are understudied and also do not perform well comparedto those from urban settings [62]. However, our findings are contrary to findings from Adeola and his colleagues found in Nigeria; that urban-based HFGCs had low participation because of a lack of political will, underfunding of misapplication of funds, weak collaboration and rivalries for power and control among participants [57].

Interestingly, public-owned PHC facilities were likely to be socially accountable by far compared to private-owned facilities. This is contrary to what has been reported from low and middle-income countries. Five studies on social accountability from Nigeria [63] and Iran [64–67] have shown private-owned healthcare facilities are more socially accountable compared to public-owned ones. However, this is not by surprise. A few years before the SRA was conducted, the government of Tanzania had conducted country-wide deliberate measures in improving social accountability among public health facilities. The measures aimed at preparing facilities for Direct Health Facility Financing mechanisms [68], the initiatives which provided autonomy on financial management at the facility level [68, 69]. We suppose that the above measures improved the situation among public facilities in Tanzania and hence more socially accountable. In recent years, there has been increasing evidence that ownership of healthcare facilities does not matter for SA [70]; therefore, the findings of this study will inform stakeholders of the position of Tanzania on this matter.

### The role of SRA in improving social accountability

As discussed above; the inclusion of five indicators in measuring social accountability in Tanzania is a great achievement for the country as most of the African studies have relied on the functionality of the facility health governing committee to describe this area. The indicators are the chosen set of approaches in the country to be followed and therefore promote clients' engagement and monitoring to improve system performance, effectiveness, and responsiveness to public needs. In health care, Quality Improvement (QI) is continuing efforts to

systematically improve the ways care is delivered to external clients (patients). SRA is a step-wise but ongoing QI initiative that aims at ensuring at least 80% of the PHC facilities become social accountable in the country by 2025.

## Limitation of the study

Assessment of social accountability in health facilities is a complex discipline. The type and number of approaches used for assessment are still debatable. As many as 37 indicators have been used in trying to assess social accountability in health facilities [29] and hence conclusion on performance becomes insufficient and comparability between studies becomes less mean-ingful. While the majority of the scholars have used a single approach to assess accountability (mostly HFGCs); our study used five indicators concurrently (health facilities committees inclusive) to increase the representativeness of the components of social accountability [13]

Again, our study did not associate the performance of individual indicators of social accountability and facility characteristics. Nevertheless, previous studies have reported that public facilities were doing better in governance mechanisms than private facilities in the pro-vision of quality care [67]

Additionally, we analysed the data that was mostly collected after document reviews at PHC facilities and from community governing offices. The relevant information (e.g. minutes) may have been forged so that the facility could get more scores during the assessment.

Moreover, the SRA used records to score the functionality of HFGC committees and other indicators. However, we believe there are circumstances whereby the committees and staff at facilities executed their roles without documenting what they did. The practice of good docu-mentation should be emphasized to communicate what has been done and properly manage the facilities.

Furthermore, the SRA tool was designed in such a way all indicators were equally important and contributed equal points to a final score of social accountability. We believe some indica-tors like the functionality of HFGCs or Boards were supposed to have more weight compared to others.

We also feel that the SRA tool should be updated to allow separate assessments on com-plaints, compliments, issues, and concerns that were conveyed to HFGC/Boards. On top of that, we would also like to see community feedback used to measure if the facility addresses local concerns or complaints rather than using external assessors. Addressing these issues will make SA results more reliable.

We excluded a high number of facilities that did not meet our inclusion criteria and this could relatively affect the strength of our study. Nevertheless, this is the first Tanzanian study on social accountability assessment having National coverage of PHC facilities. The findings will allow fair comparisons with similar studies elsewhere thus informing policymakers and health planners globally.

Finally, we did not have quality baseline data at the start of the SRA in 2015 to compare with the findings obtained in this report. Nevertheless, the findings obtained in past by other scholars show the situation was worse in Tanzania compared to now and therefore, we proba-bly associate the improvement in Social accountability among PHC facilities observed in 2017/18 and implementation of SRA since 2015/16.

## Conclusion

On average, Tanzanian PHC facilities are yet to be socially accountable and most of them did not perform in the most social accountability initiative, i.e., functionality of health facility boards or committees [71].

However, SRA initiatives could be the factor why the situation in 2018 is better compared to what was previously reported.

We recommend that the established HFB or HFGC are trained in SA mechanisms and on how to use the SRA tool in managing the PHC facilities towards achieving recommended SA status. Furthermore, health facility providers should be trained on effective data collection, use and sharing with the community. Council Health Management Teams should make sure that facilities adhere to the recommended social accountability guidelines through effective supervision and mentorship.

The SA section of the SRA tool needs to improve so that it captures feedback from the community as well on the performance of PHC facilities. Moreover, since SA is a broad discipline and it is difficult for a simple SRA tool to capture all the arguments together; we recommend further research that will explore in-depth clients' opinions on whether PHC facilities are socially accountable or not.

## Supporting information

**S1 Appendix. Clean data that was extracted from DHIS2 (https://dhis.moh.go.tz/).**
(XLSX)

## Acknowledgments

The authors are passing their sincere gratitude to the Ministry of Health, Community Development, Gender, Elderly and Children (MoHCDGEC) especially, the Health Quality Assurance Unit for permitting us to use the SRA data.

Apart from government institutions, the authors extend appreciation to development partners such as World Bank, Centres for Disease Control and Prevention (CDC), Danish International Development Agency (DANIDA), and The World Health Organization whom together made SRA possible. Others were the communities of the facilities visited, PharmAccess International, Association of Private Health Facilities in Tanzania (APHTA), Christian Social Services Commission (CSSC), and Development Partners in Health-Group (DPG-H).

## Author Contributions

**Conceptualization:** Erick S. Kinyenje, Talhiya A. Yahya, Joseph C. Hokororo, Eliudi S. Eliakimu, Mbwana M. Degeh, Omary A. Nassoro, Chrisogone C. German, Radenta P. Bahegwa, Yohanes S. Msigwa, Ruth R. Ngowi, Laura E. Marandu, Syabo M. Mwaisengela.

**Data curation:** Erick S. Kinyenje, Talhiya A. Yahya, Joseph C. Hokororo.

**Formal analysis:** Erick S. Kinyenje, Talhiya A. Yahya, Joseph C. Hokororo.

**Methodology:** Erick S. Kinyenje, Talhiya A. Yahya, Joseph C. Hokororo, Syabo M. Mwaisengela.

**Software:** Erick S. Kinyenje.

**Supervision:** Talhiya A. Yahya, Joseph C. Hokororo, Eliudi S. Eliakimu.

**Validation:** Erick S. Kinyenje, Talhiya A. Yahya, Joseph C. Hokororo, Eliudi S. Eliakimu.

**Visualization:** Eliudi S. Eliakimu.

**Writing – original draft:** Erick S. Kinyenje, Talhiya A. Yahya, Joseph C. Hokororo, Eliudi S. Eliakimu, Mohamed A. Mohamed, Mbwana M. Degeh, Omary A. Nassoro, Chrisogone C.

German, Radenta P. Bahegwa, Yohanes S. Msigwa, Ruth R. Ngowi, Laura E. Marandu, Syabo M. Mwaisengela.

**Writing – review & editing:** Erick S. Kinyenje, Talhiya A. Yahya, Joseph C. Hokororo, Eliudi S. Eliakimu, Mohamed A. Mohamed, Mbwana M. Degeh, Omary A. Nassoro, Radenta P. Bahegwa, Yohanes S. Msigwa, Ruth R. Ngowi, Laura E. Marandu, Syabo M. Mwaisengela.

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
