## [Decision Letter · Decision Letter 0]

6 Dec 2021

PONE-D-21-24574Social accountability in primary health care facilities in Tanzania: results from Star Rating AssessmentPLOS ONE

Dear Dr. Kinyenje,

Thank you for submitting your manuscript to PLOS ONE. After careful consideration, we feel that it has merit but does not fully meet PLOS ONE’s publication criteria as it currently stands. Therefore, we invite you to submit a revised version of the manuscript that addresses the points raised during the review process.

We look forward to receiving your revised manuscript.

Kind regards,

Orvalho Augusto, MD, MPH

Academic Editor

PLOS ONE

Reviewers' comments:

Reviewer's Responses to Questions

**Comments to the Author**

1. Is the manuscript technically sound, and do the data support the conclusions?

Reviewer #1: Partly

2. Has the statistical analysis been performed appropriately and rigorously? 

Reviewer #1: I Don't Know

3. Have the authors made all data underlying the findings in their manuscript fully available?

Reviewer #1: No

4. Is the manuscript presented in an intelligible fashion and written in standard English?

Reviewer #1: Yes

5. Review Comments to the Author

Reviewer #1: This is an interesting paper that describes how the government/MoH in Tanzania has taken steps to promote social accountability as an element of quality of care by using a standardized national tool. If this is consistently applied across PHCs at regular intervals it may become an important mechanism to introduce and sustain social accountability practices. The paper adds value to the existing literature as I think there is limited knowledge/reporting on this approach and process in Tanzania.

I think the objectives of the paper are interesting – to not only assess SA performance (goal) but also to explore whether the tool can be a mechanism of SA in itself (means). However, the second specific objective is not addressed in the findings or discussion section.

A substantial issue that needs to be addressed is the presentation of the SRA and the specific SA elements. Now the SRA methodology is described in multiple places (introduction, methods) but it would be clearer if it was presented in one place: including the origin of the SRA and, specifically the identification of the 5 social accountability indicators, the origin, development and application of the assessment tool (table 1). It is unclear who designed the tool, the authors or the government/MoH. It should be made clear that the SRA is a national approach, the tool existed, the data were collected prior to this study and the authors are conducting a separate analysis. Most detailed comments in attached review document are related to this issue.

It is also necessary to better explain the link between the 5 indicators, the literature review and the conceptual framework.

The arguments in some of the discussion sections need to be revised as to better reflect the findings.

6. PLOS authors have the option to publish the peer review history of their article (what does this mean?). If published, this will include your full peer review and any attached files.

Reviewer #1: **Yes: **Elsbet Lodenstein

---

## [Author Response · Author response to Decision Letter 0]

27 Jan 2022

On behalf of the co-authors, we thank E. Lodestein for her very constructive comments and guidance. This was one of the best Reviewer we have ever met. We also thank you Editor for ensuring we meet right reviewers. 

Dear editor, we humbly submit the responses to comments raised by the reviewer. Thank you again for giving us the opportunity. 

Review manuscript number PONE-D-21-24574

By: E. Lodenstein

03-12-21

Introduction

Structure.

- Start with defining what quality is and why it is problematic in TZ 

- Then the section on QI

- The social accountability (SA) element of the SRA is the focus of the study. If you decide to mention other forms of accountability as you do on the top of page 5, they need to be explained or you have to leave it out. 

Thank you for the comment to improve the structure of our manuscript. Quality has been defined and explanation on why it is a problem in Tanzania has been added. The other forms of accountability listed on top of page 5 have been deleted. All changes are shown in track changes in the revised manuscript.

I had a difficult time understanding what the SRA is, how it was developed, when it was applied (twice so far, I understand), by whom etc. Is it government initiated? Who defined the 12 domains and the assessment criteria etc. Are these criteria based on government policy (e.g. regarding the functioning of HFGC)? What was the “stepwise improvement process towards an accreditation status”? What is “pre-accreditation”? The information on the SRA is presented at different places (introduction + methods) but you could think about presenting it in one overview- perhaps a box. 

Aim of the study (page 6/7)

- The statement of the aim could be reformulated to make it more readable: e.g. This paper aims at describing the status of social accountability in public and private PHC facilities in Tanzania after the three years of implementation of Star Rating Assessment from (2015/16 to 2017/18) as part of the then broad government initiative termed “Big Results Now. 

Thank you for your comment. The sentence has been reformulated as you have suggested.

- “Describing the status” is unclear: do you mean “assessing the social accountability performance of PHC”

Thank you for the comment. This has been rephrased to “assessing the social accountability performance of public and private PHC facilities”

- “after three years of implementation” suggests you are conducting an assessment and compare results against a baseline situation. However, from the results section I see that you are assessing the performance at one point in time based on data from 2017/2018. In that case, the mentioning of “three years after implementation” is confusing. To simplify, you could just reformulate “….In Tanzania as part of the Star Rating Assessment that was conducted in 2017/2018”. 

Thank you for this observation. The sentence has been edited to make it more concise as you suggested.

Specific objectives:

- “during SRA” is unclear. I understand the SRA is just the assessment that takes place at one moment in time, it is not an intervention to improve SA. This has been deleted and hence the first specific objective has been rephrased to read as follows: “To determine proportion of PHC facilities with functional of social accountability mechanisms based on the SRA results.”

- “functional indicators” is unclear. Do you mean “PHC facilities that perform well according to performance indicators of SA”? or “PHC that have functional SA mechanisms in place”? We mean PHC facilities that have functional SA mechanisms in place. This has been edited in the specific objective number 1.

- The analysis goal mentioned on top of page 7 on performance accountability seems a third specific objective. However, that objective is not achieved in the paper. Also, the concept of “performance accountability” is not explained in the paper, so I suggest to explain it or take it out. Since you position the SA assessment in the context of Quality of Care I would stick to the concept of quality, rather than introducing another concept (performance). Thank you for this observation. The SRA had a countrywide target of all PHC facilities to improve to 3-stars and above, which very much had a nudge effect to PHC facilities staff to perform better. Therefore, the sentence has been rephrased to reflect this and the “performance accountability has need defined.

Methods

- From the introduction, I understand that there is a standard SRA tool that is being applied in the country across PHCs. I then assume that for the 5 indicators on Social Accountability, the tool already had identified indicators and measurements prior to this study. What is the origin of table 1: is it derived from the standard existing SRA tool (and how was it developed – see comment on introduction before)? OR was it developed by the researchers? 

Thank you for your comments. Table 1 originated (derived) from the SRA Tool and modified to a language of publication.

- If it was an existing tool in the context of SRA – how do the literature review and the conceptual framework link to the tool? Thank you for this observation. The literature review looked at existing literature based on the indicators of social accountability in the tool. Also, the conceptual framework was linked to the indicators, however its design was a bit disconnected to the way the data analysis was done. This has been corrected. Now all the indicators are independent variables which contribute to functional social accountability mechanisms in a PHC facility.

- The conceptual framework seems to suggest that 2 out of the 5 indicators are independent variables. However, they are not analyzed as such, can you explain that? This has been corrected. Now all the indicators are independent variables which contribute to functional social accountability mechanisms in a PHC facility.

Table 1.

- The title of the table is complex, is it not simply a “tool to assess SA performance”? 

The title is now as simple as “A tool to assess Social Accountability performance at healthcare facilities in Tanzania”

- Indicator 1 – functional facility governance committees or boards. 

o Committees can only receive 0 or 6 points (right column). But in the analysis section you state they can also receive 0,5 points. For this indicator, does it mean facilities get 0,5 points if they meet 2,3,4, or 5 characteristics? If so, it should be added in the right column that it is also possible to have a score of 2,3,4,5. 

Not all indicators had an option of Partial (0.5) scores. Only indicator number 3 had three options; Yes (1.0), Partial (0.5), and (0.0). The other 4 indicators had either a Yes or No score. We aimed to communicate this message by the sentence “Each question from the checklist had up to three responses: ‘Yes’ (score=1) or ‘No’ (score=0) or ‘Partial’ (score=0.5)”. However, since the sentence was still confusing; we restructured it to “All questions had two responses Yes’ (score=1) or ‘No’ (score=0) except the question on the “Key information on available resources is displayed” indicator that had the addition of ‘Partial’ (score=0.5)”

o Key information on available resources is displayed; the scores do not seem exclusive, e.g. “at least 2” can also be “all three”. 

Thank you for this observation. We did a typo error when we were re-writing the tool for publication. The correct sentence should have been “Partial=0.5 was given if 2 items displayed “ and not “Partial=0.5 was given if at least 2 items displayed”

Data collection section: if the data collection, cleaning and compilation is not done by the authors, this section describes not how data were collected for the study, but how the data that make up the national dataset are collected. This is an important difference that needs to be clarified. From my understanding, the authors did not collect the data and did not perform the scoring. They only performed the analysis. 

Yes, madam, the authors did not collect the data. We have restructured the data collection section to reflect our role as authors in this study and to avoid confusion. 

The description of the calculation of the scores seems complicated and can be simplified (page 11). Just explain that a facility could gain 5 points maximum and they needed 4 to be qualified as socially accountable/well performing. A suggestion: “First, the individual scores were calculated (the score for the indicator divided by the maximum possible score x 100 to give a percentage score). Secondly, the total score across the 5 indicators was determined by calculating the average of the percentage scores for the 5 indicators”. 

Thank you, madam. All that you suggested have been included in our revised document.

Moreover, after scanning thoroughly on how this outcome was calculated, we observed that there was an error in formulating the equation. We corrected and repeated analyses that are related to this variable. Therefore, please accept the changes that have occurred in the Abstract, Results, Discussion sections.

The cut-off point of 80% is described twice on page 12 – this is a repetition. 

Thank you, madam. The repetition is omitted in a revised document.

Data management and analysis (page 12). “the data were checked…” What data are you referring to? The actual data sources (minutes etc..), the scores?

We were referring to data that is kept in the national database. However, we agree with you that the statement was confusing and decided to restructure the section.

Results

- “study participants” seems the wrong term for health facilities. Rather refer to “description of health facilities under study”.

Thank you for the comment. We agree with you and adopted the suggestion

- How do you explain the high number of excluded facilities? What does it say about the methodology or the completeness of the database? (perhaps not discuss in results section but in discussion or study limitations)

The discussion about the high number of excluded facilities is now presented in the study limitation sub-section

- Reference to table 1 should be table 2 in the text

Thank you for the observation. The correction has been made.

- Page. 13. I think “functional indicators” is not the right formulation as indicators do not function, they are just representations of data. Perhaps something like “proportion of well performing health facilities regarding social accountability” or “proportion of health facilities with functional social accountability practices”. 

We thank you for the comment and agree with you. “functional indicators” has been replaced by “Proportion of health facilities with functional social accountability mechanisms”

- Figure 2 title is not correct. What you are presenting is not the proportion but the distribution of performance for different social accountability components. The description of the 5 elements on the left should then also change, leaving out the first part “proportion of facilities that”. You could also make the figure more readable by using words for the legends, e.g. yellow = good performance, orange = average performance, grey = poor performance. In the text (page 13), you state that 45,9% were found to be socially accountable. The average of data in figure 2 however, lead to a percentage of 50,48%. I may be wrong but please check this. 

Thank you, madam, this is another very important observation from you. We agree with you and changes have been made accordingly. However, there are two things to distinguish here..30.4% (previous 45.9%) are facilities that scored 4 out 5 components=socially accountable AND 50.5% is an average score from all five components (The facilities scored half of allocated 100 points for the five indicators)

- Page 14. Part of the title “Facility related characteristics…” is not necessary – you can leave out SRA of 2017/2018. 

Thank you for the comment. The part of the title has been omitted.

- Page 14. “…achieving social accountability status during assessment…” seems a complicated way of formulating, perhaps alternative: “..had increased likelihood of performing well on social accountability…”

Thank you for the comment. The sentence has been restructured

Discussion

- Page 15. The statement “…these findings are in line with….” Is very unclear. Performance of what, of whom? What does it mean “cross above the half of the allocated scores”. Which scores? You would need to give some more detail of the study you are referring to.

Thank you for the comment. The paragraph has been modified to give a clear message and the details of the studies referred to have been given. 

- The section on health facility governing committees speaks about many other elements that are not related to the 6 sub-indicators assessed under this component (e.g. non-responsiveness was a separate indicator, commitment of health care workers is again another topic, and independence of the committee was not assessed in the SRA..). I suggest you stick to your findings in the discussion section and compare with findings of similar studies.

Thank you for the good comment. Now the discussion has been restructured to focus on sub-indicators of health facility governing committees.

- The section on display of information: rather than comparing with Kenya, it may be more relevant to discuss why the display of information may have been low. Is it not a requirement from the MoH, do the facilities have this information? And if you do not know, it may be a question for further research.

Thank you for the comment. 

Yes, we were not able to find why few facilities displayed the information by using the database we have. However, we have presented what our colleagues found two years ago regarding the reasons why the display of data is poor among the facilities.

- The argument on health facilities addressing local concerns: you suggest a link between poor functioning health committees and addressing local concerns but the indicator was about facility management addressing concerns and not facility committees. Please check this argument again. Also, given your conceptual framework, could you have done a statistical analysis of this association between functionality of committees and addressing local concerns? 

The facility committees have a role to hold the facility management accountable to their performance. Therefore, if a committee does not follow up on this in their meetings, then there may be laxity in the facility management. Nevertheless, it might not be possible to associate the functionality of committees and address local concerns; they are both predictors of social accountability. 

Later on, you also advised replacing one of the sections above by a discussion on how the SA elements sits within the overall QI initiatives and the overall SRA assessment. We agreed with you and therefore decided to omit the section “health facilities addressing local concerns”. 

- Section on engagement of health providers could involve a discussion on whether engagements around public health education and campaigns constitute an element of social accountability. In my view, having meetings with communities around immunization (with evidence in minutes) does not say much about the quality of engagement and whether these meetings are opportunities for communities to call health providers to account. In literature, e.g. McCoy, there is a discussion about the public health/health education versus social accountability role.

The SRA tool assessed whether there were minutes at the community authority’s office that show the participation of health providers at the community level. This could be one of the limitations of the tool since the quality of the meeting was not clearly assessed. We have pointed out this weakness in the manuscript text. 

The section has been modified to suit the suggestions you provided.

- Page 19. This is an unclear statement, please elaborate: “Limited findings are contrary to ours whereby private-owned facilities performed better than public-owned facilities”. Which findings, from whom, about what?

Clearer details about the studies cited in the text have been given. They were all about social accountability 

- Maybe you could replace one of the sections above by a discussion on how the SA elements sits within the overall QI initiatives and the overall SRA assessment. So as to refer back to the introduction on QI and the larger SRA process. 

The section regarding “health facilities addressing local concerns” has been omitted to give a space for the section you suggested.

The role of SA elements and SRA in relationship to Tanzanian QI initiatives has been described under a section titled “The role of SA assessment in Tanzanian QI initiatives”

Limitations of the study

I think you should include limitations on the methods and tools as well. E.g. discuss implications of the following issues for your findings: tool (table 1) :

- No weighing, all indicators evenly important

- Scoring often dependent on one data source: minutes

- Complaints, issues, or concerns are different things, may be treated differently by the boards and facility managers. No distinction made.

- “Facility addressed local concerns”, seems a difficult one to assess as probably hard to link concerns to plans and vice versa. Also, it is communities who should assess this performance rather than external assessors. 

Thank you for these great inputs. We agree with you and we have included them based on the context.

Conclusion

- The first section seems to present an invalid statement – you did not compare between 2015 and 2017/2018 so you cannot make the statement that the situation improved. Also, your findings do not suggest that SRA can be a mechanism for performance accountability.

The confusing phrases “improved situation” and “mechanism for performance accountability” have been omitted.

- The recommendation is interesting but maybe you can be more specific: training on the use of SRA assessment, or on the actual use of the tool and scores (e.g. displaying them or discussing them with service users to identify improvements etc.) so that it becomes a SA dialogue tool in health facilities?

Your suggestion has been accepted and changes have been made.

- Perhaps include a reflection on the need for further research on the quality of SA/feedback process. The assessment scores are a simple representation of reality, it would be interesting to further understand the processes of feedback, and the level of satisfaction service users have with the SA mechanisms etc. 

Thank you for the reminder. An important reflection on the need for further research has been added.

---

## [Decision Letter · Decision Letter 1]

23 Feb 2022

PONE-D-21-24574R1

Social accountability in primary health care facilities in Tanzania: results from Star Rating Assessment

PLOS ONE

Dear Dr. Kinyenje,

Thank you for submitting your manuscript to PLOS ONE. After careful consideration, we feel that it has merit but does not fully meet PLOS ONE’s publication criteria as it currently stands. Therefore, we invite you to submit a revised version of the manuscript that addresses the points raised during the review process.

We look forward to receiving your revised manuscript.

Kind regards,

Orvalho Augusto, MD, MPH

Academic Editor

PLOS ONE

Additional Editor Comments:

This report has improved since the last version. However, there are still some more outstanding issues, particularly those raised by one of the reviewers below.

1. The main outcome of the analysis is an ordinal score of accountability congregating different dimensions (collected as a binary variable). The authors continue the classic approach of dichotomization of such kind of score. It is OK but that has the effect of throwing away the ordinal information among those below the cut-off.

2. Please do not just report p-values and the point estimate of odds-ratio (OR) only. Please report the 95% confidence interval of OR.

3. Table 3:

- Call it multivariable (not multivariate)

- Add below the table (as a footnote) what variables were used to adjust for.

4. Stata is not an acronym. Please write Stata not STATA. And please add a citation.

Reviewers' comments:

Reviewer's Responses to Questions

**Comments to the Author**

1. If the authors have adequately addressed your comments raised in a previous round of review and you feel that this manuscript is now acceptable for publication, you may indicate that here to bypass the “Comments to the Author” section, enter your conflict of interest statement in the “Confidential to Editor” section, and submit your "Accept" recommendation.

Reviewer #1: All comments have been addressed

Reviewer #2: (No Response)

2. Is the manuscript technically sound, and do the data support the conclusions?

Reviewer #1: Yes

Reviewer #2: Partly

3. Has the statistical analysis been performed appropriately and rigorously? 

Reviewer #1: I Don't Know

Reviewer #2: Yes

4. Have the authors made all data underlying the findings in their manuscript fully available?

Reviewer #1: Yes

Reviewer #2: No

5. Is the manuscript presented in an intelligible fashion and written in standard English?

Reviewer #1: No

Reviewer #2: Yes

6. Review Comments to the Author

Reviewer #1: Review of revision 1

Elsbet Lodenstein

14 feb. 2022

The authors have made very important revisions. Comments have been adequately addressed. The clarifications and changes in the calculation make the study more consistent and trustworthy and the discussion is relevant and interesting.

Revision needed in Methods section:

- The authors need to explain how table 1 was constructed just as they do in the responses to the reviewer. So, basis is SRA tool but complemented with criteria from literature. It needs to be described explicitly which elements are from the original SRA tool and which elements were added by the authors.

- In the responses to reviewer, the authors state that “Now all the indicators are independent variables which contribute to functional social accountability mechanisms in a PHC facility”. However, in the text, under study variables this was not yet adapted.

Other minor revisions:

- Page 3 bottom – Big Results Now initiative needs to be referenced.

- Top page 5: definitions of SA need a reference.

- Page 5, listing of 5 elements assessed (also mention that it is five elements) – formulation can be simplified and made into nouns. E.g. healthcare workers engagement with the local community; facility addressing local concerns; community participation in facility planning process etc…

- The “Hence….” sentence could be taken out as objective of the paper is well explained below on page 7.

- Page 7. Aim statement improved but I think it is still complicated. ….”SRA Tools to assist as a mechanism for making facility in-charges and other staff accountable in ensuring good performance of their facility in terms of providing quality services”. Why not just …”SRA Tools to assist as a mechanism for making facility in-charges and other staff accountable for providing quality services”. And leave the performance out. I would also exclude the concept of performance accountability because it is again confusing as the paper focuses simply on performance of social accountability, not performance in terms of agreed targets.

Results

- the sentence on page 14 “The average score in percentages….” should be explained more clearly. Something like “This means that facilities’ overall score for performance of social accountability across the 5 indicators was 50,5%.

I recommend to have an English editor conduct proof reading. Overall the text is well written, but the proofreading would take out some omissions, like

- missing articles (e.g. “a” is missing before “variety” on top of page 4; “the” is missing before “development on page 6 or “The” is missing before “Composition” on page 6; there are many similar small mistakes).

- Formulations: e.g. p. 12 sentence does not run smoothly. “Missing data were not imputed and therefore excluded the in analysis”.

Reviewer #2: The authors have revised a paper describing the results of the Social Accountability measures from the 2018=2019 survey done part of the Tanzania Star Rating Assessment. The paper is important in highlighting the importance o this component of facility performance and they should be applauded for the work. They have also worked hard to respond to another reviewers comments, which have improved the readability and strength of the paper. I however still struggled with a number of areas and some of the writing needs to be reviewed. In particular, while they re clear in their response that this is a single cross sectional report in their responses, there are still areas where the language implies change over time (“how star rating improved Social accountability \\” in the introduction and describing how areas were affected when they are only associated in the discussion.

Introduction

While there is an interesting introduction, I got a bit lost in the discussion about accountability (is it the country which is held accountable to existing and emerging social concerns or in this area-the facility and health system?

In the description of how the HFGC is formed, what does “transparent manner at respective levels” mean?

I was still confused about the introduction of a conceptual model on page 6 (Molyneux) and Lodensteins definition of social accountability. Is this about the reasons why these structures are valid? And so why they should be a measure of social accountability?

I am also a bit confused about the aims-particularly “the analysis aims at showing the potential of the SRA tools to assist as a mechanism for making facility in-charges and other staff accountable….” I did not see that in results or discussion?

Methods

Table 1 I assume is not just a tool, but actual the section from the STAR assessment? Consider adding that

Why is a 2 point assessment discussed when only one assessment is used (would be very interesting to see change over time)

How were the independent factors chosen? Were there others (such s overall Star rating?)

The use of the term performance scores is a little confusing-do you mean the SA scores?

Results:

• I think there is a missing space in the first subtitle “understudy:?

• Importantly-why did 58.4% not meet eligibility-would consider a consort diagram f they fell out by different reasons. This is also a major potential bias?

• It would be helpful to also see the distribution of the results which make up the Functional facility governance committee or boards. What were the areas which resulted in 0? Even a description of the results from 0-6 would be of great interest. Some seem a bit subjective “adequately trained”.

• Can the authors include the actual tool and any scoring information in terms of how the data were collected. If there were no minutes from committees in the last 6 months-how was that scored as an example but applies to others as depended on documentation

• In the figure-how can you have overall ratings when ratings were missing from some of the components of the 5 areas? Also the figure (on my print out) seems a bit blurry

Discussion

As noted above, you do not know if the determinants affected the rating-more association

The second paragraph is a bit repetitive and describe the methods and results. It would be better to dive into the discussion of the results

There is a sentence alone “experience has shown the existence of poor social accountability mechanisms among health facilities in Tanzania-whose experience and needs expansion (and references. How does this support or not your findings

For the part of the HFGC-which components gave the most challenge? Are there any references showing the impact of well run committees?. The statement about HFGCs being ineffective also needs a bit more detail and again if and how similar to or different from results

The statement about use of records to score (which I think is important) however may belong better in limitations unless you are discussing the instrument itself (versus study)

The section on displayed information is interesting, however the discussion about data analysis does not seem to be as relevant as the displayed information was budgets and resources?

Health work engagement: The authors are correct that the way the indicators was assessed “could have exaggerated the findings”-but wonder if that is true for most or all of the areas measured?

The use of the term “forced” in convening needs explanation-how were they forced?

Engaging community: The statement about hampering by gaps in manpower, finance and infrastructure is important and could use more detail in terms of what is needed to address the gaps as measured by the STAR SA tool

The section on the role of SRA n improving social accountability was a bit hard to understand as no information as in the results about how these results were used. I think potentially one of the most important potential new insights and would be helpful to know progress particularly now that we are in 2022

The authors have done a through job around limitations (see however comment above). However the important statement about updating the SAR is not a part of limitations but perhaps goes netter into a discussion about the tool? Or conclusions and next steps?

As noted above-the authors need to describe the exclusion criteria (and I think that is a typo as they may have meant ‘did not meet our inclusion criteria)

Writing

• The authors have some extremely long sentences which make it hard to a reader to understand the main points. As examples, the end of the very long paragraph in the introduction starting with “to achieve”

• All acronyms need spelling out (like the first time SRA is used) and would decrease use of ones which are not common (ex HFGC/HFB).

• All quotes should be identified for their source-this is missing in a number of places.

7. PLOS authors have the option to publish the peer review history of their article (what does this mean?). If published, this will include your full peer review and any attached files.

Reviewer #1: **Yes: **Elsbet Lodenstein

Reviewer #2: No

---

## [Author Response · Author response to Decision Letter 1]

16 Apr 2022

Dear editor, we humbly submit the responses to comments raised by the reviewers including you. Thank you again for giving us the opportunity. 

Additional Editor Comments:

This report has improved since the last version. However, there are still some more outstanding issues, particularly those raised by one of the reviewers below.

1. The main outcome of the analysis is an ordinal score of accountability congregating different dimensions (collected as a binary variable). The authors continue the classic approach of dichotomization of such kind of score. It is OK but that has the effect of throwing away the ordinal information among those below the cut-off.

Dear Editor, we agree with you that this is one of the great challenges for binary logistic regression. Thank you for the reminder. 

2. Please do not just report p-values and the point estimate of odds-ratio (OR) only. Please report the 95% confidence interval of OR.

Thank you for your comment. The 95%CI of OR are now included.

3. Table 3:

- Call it multivariable (not multivariate)

The change has been adapted.

- Add below the table (as a footnote) what variables were used to adjust for.

We have added the footnote explaining which variables have been adjusted for.

4. Stata is not an acronym. Please write Stata not STATA. And please add a citation.

The change has been done and a citation is provided.

Reviewers' comments:

Reviewer's Responses to Questions

Reviewer #1: Review of revision 1

Elsbet Lodenstein

14 feb. 2022

The authors have made very important revisions. Comments have been adequately addressed. The clarifications and changes in the calculation make the study more consistent and trustworthy and the discussion is relevant and interesting.

Thank you for the compliment 

Revision needed in Methods section:

- The authors need to explain how table 1 was constructed just as they do in the responses to the reviewer. So, basis is SRA tool but complemented with criteria from literature. It needs to be described explicitly which elements are from the original SRA tool and which elements were added by the authors.

The statement on the originality of Table 1 has been added to the manuscript. Table.1 was derived from the SRA Tool and modified to a language of publication, however, none of the indicators was changed.

- In the responses to reviewer, the authors state that “Now all the indicators are independent variables which contribute to functional social accountability mechanisms in a PHC facility”. However, in the text, under study variables this was not yet adapted.

Thank you for noting this dear Reviewer. The sentences have been modified to adapt to the changes. 

Other minor revisions:

- Page 3 bottom – Big Results Now initiative needs to be referenced.

The reference has been added.

- Top page 5: definitions of SA need a reference.

The reference has been added.

- Page 5, listing of 5 elements assessed (also mention that it is five elements) – formulation can be simplified and made into nouns. E.g. healthcare workers engagement with the local community; facility addressing local concerns; community participation in facility planning process etc…

The phrase “five indicators” has been added. The formulation had been made to nouns as suggested.

- The “Hence….” sentence could be taken out as objective of the paper is well explained below on page 7.

The sentence has been taken out.

- Page 7. Aim statement improved but I think it is still complicated. ….”SRA Tools to assist as a mechanism for making facility in-charges and other staff accountable in ensuring good performance of their facility in terms of providing quality services”. Why not just …”SRA Tools to assist as a mechanism for making facility in-charges and other staff accountable for providing quality services”. And leave the performance out. I would also exclude the concept of performance accountability because it is again confusing as the paper focuses simply on performance of social accountability, not performance in terms of agreed targets.

Your suggestion on simplifying the sentence has been adopted and the concept of performance accountability has been excluded as well to avoid confusion. 

Results

- the sentence on page 14 “The average score in percentages….” should be explained more clearly. Something like “This means that facilities’ overall score for performance of social accountability across the 5 indicators was 50,5%.

Thank you, your suggestion has been adopted in manuscript text. 

I recommend to have an English editor conduct proof reading. Overall the text is well written, but the proofreading would take out some omissions, like

- missing articles (e.g. “a” is missing before “variety” on top of page 4; “the” is missing before “development on page 6 or “The” is missing before “Composition” on page 6; there are many similar small mistakes).

Thank you, I have worked on it.

- Formulations: e.g. p. 12 sentence does not run smoothly. “Missing data were not imputed and therefore excluded the in analysis”.

The sentence has been reconstructed.

Reviewer #2: The authors have revised a paper describing the results of the Social Accountability measures from the 2018=2019 survey done part of the Tanzania Star Rating Assessment. The paper is important in highlighting the importance o this component of facility performance and they should be applauded for the work. They have also worked hard to respond to another reviewers comments, which have improved the readability and strength of the paper. I however still struggled with a number of areas and some of the writing needs to be reviewed. In particular, while they re clear in their response that this is a single cross sectional report in their responses, there are still areas where the language implies change over time (“how star rating improved Social accountability \\” in the introduction and describing how areas were affected when they are only associated in the discussion.

Thank you for the constructive comment and compliment. In any case, when the word “improved” is mentioned throughout the document, it is for the purpose of showing the situation has changed when compared to findings from previous studies. The references are cited in the text. We did not compare the baseline and re-assessment findings. 

Introduction

While there is an interesting introduction, I got a bit lost in the discussion about accountability (is it the country which is held accountable to existing and emerging social concerns or in this area-the facility and health system?

We got the reviewer’s concern. We have improved the confusing sentence from “In the context of PHC, social accountability is a measure of whether a country is held accountable to existing and emerging social concerns and priorities based on need” to “In the context of PHC, social accountability is a measure of whether a country and especially the health facility, are held accountable to existing and emerging social concerns and priorities based on need”

In the description of how the HFGC is formed, what does “transparent manner at respective levels” mean?

“transparent manner at respective levels” means the members were publicly selected to represent the community at the level of their authorities i.e. Hospital Advisory Body members would represent a community that is served by council hospital (about 250,000 population) while members of HFGC at the Dispensary level would represent the community at specific catchment area of about 10,000 population. The sentence has been rephrased to be more understandable.

I was still confused about the introduction of a conceptual model on page 6 (Molyneux) and Lodensteins definition of social accountability. Is this about the reasons why these structures are valid? And so why they should be a measure of social accountability?

The definitions by these experts have just happened to be in-line with how Tanzania conceptualized on indicators needed to assess social accountability at primary health facilities. However, as explained in limitation section of the manuscript text; we expect the SRA tool will be modified in next days to incorporate views from more scholars and practioners. 

I am also a bit confused about the aims-particularly “the analysis aims at showing the potential of the SRA tools to assist as a mechanism for making facility in-charges and other staff accountable….” I did not see that in results or discussion?

Thank you for the good comment. Since the SRA tool included an assessment on whether the facility in-charges and other staff were socially accountable; the findings from the assessment will highlight the areas that need improvement from which facilities/ministries can work upon. The sentence has been modified to make it clearer. 

Methods

Table 1 I assume is not just a tool, but actual the section from the STAR assessment? Consider adding that

Your comment is well adapted. We have changed the title from “A tool assess Social Accountability Performance at Healthcare Facilities in Tanzania” to “A section on SRA tool assessing Social Accountability Performance at Healthcare Facilities in Tanzania”

Why is a 2 point assessment discussed when only one assessment is used (would be very interesting to see change over time)

Referring to the above responses, we did not compare the baseline and re-assessment findings and therefore whenever the word improved used; it was compared to past findings in the country. 

How were the independent factors chosen? Were there others (such s overall Star rating?)

We could add more variables, however, we were limited to very few which are in the current database. This has been included in the limitations of the study.

The use of the term performance scores is a little confusing-do you mean the SA scores?

The sentence has been modified to avoid confusion.

Results:

• I think there is a missing space in the first subtitle “understudy:?

Thank you for the suggestion. However, to the best of our understanding and if you agree with us-we thought it should remain as “understudy” and not “under study”. Or else, because the word itself brings some confusion.. we have opted to change the title to “Description of participating health facilities” 

• Importantly-why did 58.4% not meet eligibility-would consider a consort diagram f they fell out by different reasons. This is also a major potential bias?

Yes, this was one of the major limitations of the study. We excluded a high number of facilities that did not meet our exclusion criteria and this could relatively affect the strength of our study. Nevertheless, this is the first Tanzanian study on social accountability assessment having National coverage of PHC facilities. The facilities were randomly excluded (not systematic error) and therefore less chance to cause selection bias

• It would be helpful to also see the distribution of the results which make up the Functional facility governance committee or boards. What were the areas which resulted in 0? Even a description of the results from 0-6 would be of great interest. Some seem a bit subjective “adequately trained”.

We agree with you. However, this was not in the scope of study provided the dataset prepared. We look forward to meeting your wish which is also ours. Thank you.

• Can the authors include the actual tool and any scoring information in terms of how the data were collected. If there were no minutes from committees in the last 6 months-how was that scored as an example but applies to others as depended on documentation

The data collection tool for social accountability area is attached as Table 1. The scoring was done at the indicator level. The facility needed to get all 6 verification questions right, for example; to score YES for the indicator “Functional facility governance committees or boards”. 

• In the figure-how can you have overall ratings when ratings were missing from some of the components of the 5 areas? Also the figure (on my print out) seems a bit blurry

The overall ratings were the facilities’ average scores that had no missing values. We tried to improve the clarity in this version.

Discussion

As noted above, you do not know if the determinants affected the rating-more association

The second paragraph is a bit repetitive and describe the methods and results. It would be better to dive into the discussion of the results

Thank you for the comment. The second paragraph has been omitted in this version.

There is a sentence alone “experience has shown the existence of poor social accountability mechanisms among health facilities in Tanzania-whose experience and needs expansion (and references. How does this support or not your findings.

The sentence has been modified to suit the reviewer’s recommendation. 

For the part of the HFGC-which components gave the most challenge? 

It was impossible to discuss at the question level because the data were collected and analysed at the indicator level.

Are there any references showing the impact of well run committees?. The statement about HFGCs being ineffective also needs a bit more detail and again if and how similar to or different from results

Some sentences and references have been added to show evidence of some impact of well-run committees and also to detail infective HFGCs

The statement about use of records to score (which I think is important) however may belong better in limitations unless you are discussing the instrument itself (versus study)

Thank you for this observation. The paragraph has been improved and thereafter transferred to Limitation section. 

The section on displayed information is interesting, however the discussion about data analysis does not seem to be as relevant as the displayed information was budgets and resources?

I agree with you, however, information on “plans and budget, allocation to medicines and supplies, revenue collection, received funds, and expenditure” sometimes would require analysis of data first to get the information to display on the walls. E.g. budget summary, funds and expenditures would require arithmetic calculations

Health work engagement: The authors are correct that the way the indicators was assessed “could have exaggerated the findings”-but wonder if that is true for most or all of the areas measured?

We agree with you that this could be true for other indicators and not for “Health workers’ engagement with the local community” only. Therefore, we have modified the paragraph to reflect what we wanted to communicate. We needed readers to understand that high performance in community engagement could be associated with the country’s high achievement in community healthcare outreach services.

The use of the term “forced” in convening needs explanation-how were they forced?

The more appropriate phrase has been applied in place of the “forced”

Engaging community: The statement about hampering by gaps in manpower, finance and infrastructure is important and could use more detail in terms of what is needed to address the gaps as measured by the STAR SA tool

The details have been added in manuscript text

The section on the role of SRA n improving social accountability was a bit hard to understand as no information as in the results about how these results were used. I think potentially one of the most important potential new insights and would be helpful to know progress particularly now that we are in 2022

In this section, we thought it was important to let the reader understand how the SA elements sit within the overall QI initiatives and the overall SRA assessment. So as to refer back to the introduction on QI and the larger SRA process. This was also the recommendation from one of the reviewers

The authors have done a through job around limitations (see however comment above). However the important statement about updating the SAR is not a part of limitations but perhaps goes netter into a discussion about the tool? Or conclusions and next steps?

Thank you for your comments which we have also included in this new edition

As noted above-the authors need to describe the exclusion criteria (and I think that is a typo as they may have meant ‘did not meet our inclusion criteria)

If I got you right, then we have included the exclusion criteria under the methodology part of the manuscript. 

Writing

• The authors have some extremely long sentences which make it hard to a reader to understand the main points. As examples, the end of the very long paragraph in the introduction starting with “to achieve”

We have restructured long sentences but retained the message. Thank you for the comment.

• All acronyms need spelling out (like the first time SRA is used) and would decrease use of ones which are not common (ex HFGC/HFB).

Thank you for the comment. We have cross checked the acronyms and worked on them.

• All quotes should be identified for their source-this is missing in a number of places.

We have added some references to quotes that had not indicated the sources. There was a sentence in the data extraction and management section in which we incorrectly marked quotation; We have removed its quotations.

---

## [Decision Letter · Decision Letter 2]

29 Apr 2022

Social accountability in primary health care facilities in Tanzania: results from Star Rating Assessment

PONE-D-21-24574R2

Dear Dr. Kinyenje,

We’re pleased to inform you that your manuscript has been judged scientifically suitable for publication and will be formally accepted for publication once it meets all outstanding technical requirements.

Kind regards,

Orvalho Augusto, MD, MPH

Academic Editor

PLOS ONE

Additional Editor Comments (optional):

Reviewers' comments:

Reviewer's Responses to Questions

**Comments to the Author**

1. If the authors have adequately addressed your comments raised in a previous round of review and you feel that this manuscript is now acceptable for publication, you may indicate that here to bypass the “Comments to the Author” section, enter your conflict of interest statement in the “Confidential to Editor” section, and submit your "Accept" recommendation.

Reviewer #2: (No Response)

2. Is the manuscript technically sound, and do the data support the conclusions?

Reviewer #2: Yes

3. Has the statistical analysis been performed appropriately and rigorously? 

Reviewer #2: Yes

4. Have the authors made all data underlying the findings in their manuscript fully available?

Reviewer #2: Yes

5. Is the manuscript presented in an intelligible fashion and written in standard English?

Reviewer #2: Yes

6. Review Comments to the Author

Reviewer #2: The authors have done a very careful job in responding to the additional comments. there was one comment which was not addressed and which I think are important to clarify

1.The comment was: Importantly-why did 58.4% not meet eligibility-would consider a consort diagram f

they fell out by different reasons. This is also a major potential bias?

The response:

Yes, this was one of the major limitations of the study. We excluded a high number of

facilities that did not meet our exclusion criteria and this could relatively affect the

strength of our study. Nevertheless, this is the first Tanzanian study on social

accountability assessment having National coverage of PHC facilities. The facilities

were randomly

Is confusing. I think the paper needs a consort diagram to explain which fell out and why or explain why this is not possible which was recommended in other comments or a clear explanation why this is either not feasible or not needed

7. PLOS authors have the option to publish the peer review history of their article (what does this mean?). If published, this will include your full peer review and any attached files.

Reviewer #2: No

---

## [Editor Report · Acceptance letter]

14 Jul 2022

PONE-D-21-24574R2 

Social accountability in primary health care facilities in Tanzania: results from Star Rating Assessment 

Dear Dr. Kinyenje:

I'm pleased to inform you that your manuscript has been deemed suitable for publication in PLOS ONE. Congratulations! Your manuscript is now with our production department. 

Kind regards, 

on behalf of

Dr. Orvalho Augusto 

Academic Editor

PLOS ONE